# Can Wearable Inertial Measurement Units Be Used to Measure Sleep Biomechanics? Establishing Initial Feasibility and Validity

**DOI:** 10.3390/biomimetics8010002

**Published:** 2022-12-21

**Authors:** Nicholas Buckley, Paul Davey, Lynn Jensen, Kevin Baptist, Bas Jansen, Amity Campbell, Jenny Downs

**Affiliations:** 1Curtin School of Allied Health, Curtin University, Perth, WA 6102, Australia; 2Telethon Kids Institute, Child Disability Team, Perth 6009, Australia; 3Perth Children’s Hospital, Perth 6009, Australia; 4Ace Therapy Services, Perth 6021, Australia; 5Postural Care Australia, Perth 6024, Australia

**Keywords:** Inertial Measurement Units, feasibility, validity, sleep, biomechanics

## Abstract

Wearable motion sensors, specifically, Inertial Measurement Units, are useful tools for the assessment of orientation and movement during sleep. The DOTs platform (Xsens, Enschede, The Netherlands) has shown promise for this purpose. This pilot study aimed to assess its feasibility and validity for recording sleep biomechanics. Feasibility was assessed using four metrics: Drift, Battery Life, Reliability of Recording, and Participant Comfort. Each metric was rated as Stop (least successful), Continue But Modify Protocol, Continue But Monitor Closely, or Continue Without Modifications (most successful). A convenience sample of ten adults slept for one night with a DOT unit attached to their sternum, abdomen, and left and right legs. A survey was administered the following day to assess participant comfort wearing the DOTs. A subset of five participants underwent a single evaluation in a Vicon (Oxford Metrics, Oxford, UK) motion analysis lab to assess XSENS DOTs’ validity. With the two systems recording simultaneously, participants were prompted through a series of movements intended to mimic typical sleep biomechanics (rolling over in lying), and the outputs of both systems were compared to assess the level of agreement. The DOT platform performed well on all metrics, with Drift, Battery Life, and Recording Reliability being rated as Continue Without Modifications. Participant Comfort was rated as Continue But Monitor Closely. The DOT Platform demonstrated an extremely high level of agreement with the Vicon motion analysis lab (difference of <0.025°). Using the Xsens DOT platform to assess sleep biomechanics is feasible and valid in adult populations. Future studies should further investigate the feasibility of using this data capture method for extended periods (e.g., multiple days) and in other groups (e.g., paediatric populations).

## 1. Introduction 

Wearable accelerometers have long been used in research to record body segment orientation and motion [1,2,3]. Over the past decade, Inertial Measurement Units (IMUs) have become a research tool of choice for field-based quantification of kinematics and joint angles [4,5]. By combining the outputs of the gyroscope, accelerometer, and magnetometer componentry, IMUs are able to capture dynamic movement data accurately and reliably [6]. Small IMUs worn on body segments (that do not overly encumber the individual) have been used in tasks that otherwise would have been affected by the weight and/or bulk of large wearable sensors, such as upper limb tasks [7] and gait analysis [8]. 

Using IMUs to measure body orientation and movement during sleep (collectively, sleep biomechanics) is an emerging area of interest [9,10]. Sleep biomechanics are of clinical relevance in a number of conditions. For instance, in low back pain [11], obstructive sleep apnoea [12], and Parkinson’s disease [13], the relationship between patient symptoms and certain patterns of movement and positions during sleep have been investigated. An area where the measurement of sleep biomechanics is particularly important is understanding the development of body shape distortion in people with severe physical disabilities [9]. First proposed by Fulford and Brown [14], the proposed pathogenesis is that persons with a severe physical disability have a reduced ability to vary their position during sleep. Uneven exposure to gravity over the course of years leads to the asymmetrical laying down of bone, and eventually to the development of body shape distortions such as ribcage asymmetry and scoliosis [15].

Wireless, Bluetooth-connected IMUs can be worn discreetly in the typical sleep setting and could be well suited to the purpose of measuring sleep biomechanics. Other measures, such as overnight videography [16], often necessitate participants attending a foreign sleep environment (such as a sleep laboratory) and relying on subjective assessment of sleep position, which is vulnerable to rater error. An objective measure of sleep biomechanics that can be introduced to a participant’s native sleep environment with minimal disruption therefore would have useful application as both a research and clinical tool; IMUs have the potential to be the solution for this unmet need.

Wearable position sensors have previously been successfully used to measure sleep biomechanics in adults [17], children [18], and the elderly [10]. However, most applications use a single IMU unit, which yields a single orientation [19], rendering the lying participant as a ‘barrel’ or ‘log’ polygon without any segmentation of the trunk, pelvis, and limbs. This is of limited research and clinical value, as one is unable to evaluate the movement of body segments in reference to one another (for example, trunk rotation relative to leg position). While some studies have used multiple sensors to assess sleep biomechanics (e.g., head and trunk [20]), the implementation of these systems is often highly technical and is not practical for clinical or participant use [21].

With the increased availability of economic, smartphone-operated sets of IMUs (such as the Xsens DOT platform, abbreviated to DOTs), the development of a clinically applicable assessment tool for sleep biomechanics is timely. However, as IMUs have not been used for this purpose before, a number of technical and practical considerations need to be evaluated prior to further research and development. The first aim of this study was to examine four main feasibility metrics: drift (slow change of orientation signal independent of the measured parameter, which can induce error), battery life (to record a full night of sleep biomechanics data), reliability of recording (recording continuously with minimal gaps in data), and participant comfort (whether wearing the DOTs were well tolerated and did not unduly disturb sleep). 

Secondly, this study aimed to establish the validity of using the Xsens DOT platform to measure sleep biomechanics by comparing the output to that of a known motion capture gold standard, a Vicon Motion Analysis Laboratory [22,23,24], to ensure confidence that recorded sleep biomechanical orientation data captured using the DOTs alone in future studies can be trusted.

## 2. Materials and Methods

### 2.1. Ethics 

This study received approval from Curtin University’s Human Research Ethics Committee (approval number HRE2020-0138).

### 2.2. Study Design

Feasibility was examined through a pilot study, specifically examining technical and practical aspects of using the Xsens DOT platform (Enschede, The Netherlands) for the examination of sleep biomechanics. This study followed the framework suggested by Thabane et al. [25]; as recommended, questions were stated a priori and the results rated against specific criteria (Table 1). The questions we aimed to answer were:Do the Xsens DOTs remain accurate with minimal drift over an entire night?Do the Xsens DOTs have sufficient battery life to record an entire night of sleep?Do the Xsens DOTs reliably record data with minimal signal dropout?Are participants comfortable while sleeping when wearing the Xsens DOTs?

Following data analysis, each endpoint was then given one of the four ratings as per Thabane et al.: Stop (least successful), Continue But Modify Protocol, Continue But Monitor Closely, or Continue Without Modifications (most successful). Details of the feasibility endpoints for this study are presented in Table 1.

A psychometric study was conducted to assess validity, comparing the level of agreement between the accurate Motion Analysis Laboratory and the Xsens DOT platform. Both systems simultaneously recorded simulated sleep movements and positions, and the level of agreement between the two measures was assessed.

### 2.3. Participants

Ten adult volunteers were recruited via convenience sampling. Participants were included if they were 18–60 years old and generally fit and healthy. Adults were included in this study as compliance with study protocol was of critical importance. Participants were excluded if they had acute or chronic injuries or pain (e.g., acute or chronic lower back pain) that may have limited mobility and comfort while sleeping. A subgroup of 5 participants took part in the validity study in the Motion Analysis Laboratory testing.

### 2.4. Materials

Four Xsens DOT sensors (Enschede, The Netherlands) (Version 2.0, 4 Hz sample, ±2000 deg/s gyroscope, ±16 g accelerometer, ±8 Gauss magnetometer) were used, paired to and programmed by a Samsung Galaxy S9 Smartphone (Samsung, Seoul, South Korea) running the native Xsens DOT application (Version 2.0, Xsens, Enschede, The Netherlands). The DOTs were attached to the participant’s sternum, anterior abdomen, and left and right distal thighs (see Figure 1) using 100 mm × 100 mm pieces of Tegaderm adhesive dressing applied over the sensors (3M, Saint Paul, MN, USA). This was preceded by a 10–15 min trial of a small segment of Tegaderm being placed on the ventral forearm to test for allergies.

### 2.5. Drift

Using a hexagonal box, 5 DOTs were attached to the interior facets using Tegaderm (Figure 2) as per overnight testing protocol. Once the DOTs started recording, the box was rotated clockwise one facet (i.e., 60 degrees) at 30-min intervals for a total of 6 h. At the conclusion of the test, the output of the DOTs (in Euler angles) was graphed to compare their output to the expected 60-degree changes. Of particular importance is the yaw (X-angle), as this corresponds to transverse rotation, the major movement being measured when examining sleep biomechanics.

A separate stationary test was undertaken to assess for false positive readings (i.e., recording movement when none occurred). This was completed with a DOT being left to record stationary on a flat surface and left to record for a full battery charge (approx. 8 h). The output of the DOTs (in Euler angles) was then graphed to assess for any drift.

The yaw (X-angle) component of the Euler angle output by the DOTs remaining accurate (within 1 degree of expected range) for >95% of the trial was considered as being feasible.

### 2.6. Battery Life

The projected battery life of the V2 Xsens DOT units is 8–10 h. Battery life was assessed by leaving a single DOT unit to record from full charge until the battery was fully depleted—this was repeated 10 times, each time with a different DOT unit. Following recording, the final timestamp was examined to assess how long each set of DOTs recorded data. A battery life of at least 8 h per trial was considered as being feasible, as it is within the expected range of battery life of V2 Xsens DOT units and also within the majority of the recommended 8–10 h duration of sleep for an adult [26].

### 2.7. Reliablility of Recording

The Xsens DOT supports two modes of data collection, continuous Bluetooth streaming, and Recording mode, in which orientation data are logged directly to the internal memory storage of each DOT without the need for continuous connection to the smartphone. We elected to utilise Recording mode, as Bluetooth had proved unreliable for multi-hour recording in early pilot testing—packet loss being a common issue with real-time Bluetooth streaming [27].

Running firmware v2.0 and using Recording mode, the DOTs have a variable logging rate (1/4/10/12/15/20/30/60/120 Hz). We elected to record at 4 Hz as this was a good compromise between data richness and ensuring that the recorded file sizes were not so large that multiple nights of recording would not be possible. Similar to battery life, the reliability of recording was assessed by leaving a single DOT unit to record from full charge until the battery was fully depleted—this was repeated 10 times, each time with a different DOT unit. Further, 4 Hz corresponds to an orientation being generated and recorded every 0.25 s. Drop out was therefore assessed by calculating the interval between recorded orientations and if drop out did occur, considering the length of the loss of data and how this might impact the overall interpretation of the data. As movement during sleep is of relatively low frequency and amplitude [28], a threshold of less than 5 s dropout (for <5% of the total recording time) was considered as being feasible.

### 2.8. Comfort of Participants

To test participant comfort, ten adult volunteers each wore the DOT array (chest, pelvis, left leg, right leg) for a single night in their native sleeping environment (own bed and bedroom). Prior to the completion of the overnight data collection, a General Questionnaire regarding overall health and sleep habits (e.g., demographics, bed and bedroom description, medical conditions, and medications taken; Appendix A part 1) was administered. Questionnaires were completed by each participant on REDCap^TM^, a secure, web-based software platform designed to support data capture for research studies [29]. An investigator (NB) attended the participant’s home, provided study materials, and trained the participant in the operation of the Xsens DOT app on the provided smartphone and attachment of the Xsens DOT units. The investigator then departed, and the operation of the DOTs and smartphone were fully completed by the participant overnight. The participant attached the DOTs to their sternum, anterior abdomen, and left and right distal thighs using the provided Tegaderm dressings. Participants were instructed to place the DOTs on the relevant body part in the midline, with the individual DOT facing outwards and the right way up—the exact placement of the DOT and its securing dressing was left to the discretion and comfort of the participant. After starting the recording of the devices using the app on the smartphone, they then went to bed and slept as normal. Upon waking, participants removed the DOTs, and study materials were collected by an investigator the following day.

The comfort of participants was assessed using a Satisfaction Survey to capture participant feedback and record their comfort while sleeping with the DOTs in place, as well as equipment and app ease of use. An overall rating of >70% on each of the scales by 90% of the participants was considered to indicate a feasible threshold of satisfaction.

### 2.9. Sensor Validity

A subgroup of 5 participants completed a single assessment session in the motion analysis laboratory at the Curtin School of Allied Health by investigators trained and familiar with the use of both the VICON and DOTs systems. Further, 18 MX and T-series Vicon cameras (Oxford Metrics, Oxford, UK) were used, sampling at 250 Hz, with retro-reflective trajectory data labelled, filtered, and modelled in Vicon Nexus 2.10.3 software (Oxford Metrics, Oxford, UK). DOTs, sampling at 60 Hz, were attached in the locations described above and also fitted with a 6.4 mm reflective marker (B & L Engineering, Santa Ana, CA, USA) in each corner of the DOT using double-sided tape (four reflectors per DOT). Participants were verbally prompted through a series of movements intended to mimic typical sleep biomechanics; the starting position was lying supine on a plinth. With both the VICON and DOTs systems simultaneously recording, three discrete movements were completed. First, participants were asked to roll from supine into left side lying, back to supine, and then into right side lying, holding each position for a count of 3 s (Whole Body Rolling). Second, participants were then asked to lie supine and (keeping shoulders on the plinth) rotate their knees to the left, back to midline, then to the right, holding each end range position for 3 s (Leg Drops). Finally, participants were asked to start supine, transition to long sitting, then transition back to supine, holding each position for 3 s (Sitting Up). Each of these cycles were repeated 5 times.

Both data sets were recorded in quaternions. A custom Labview program (National Instruments, Austin, TX, USA) was used to up-sample Xsens DOTs data from 60 Hz to 250 Hz to match VICON Data capture rate. The DOT platform data were then rotated 182.5 clockwise degrees around the vertical axis to match the VICON Motion Analysis lab coordinate system. Temporal synchronisation of the two data sets was then completed via cross-correlation of orientation between data sets.

In STATA 16 (StataCorp, TX, USA), a mixed model was used to examine the level of agreement between the orientation (in degrees) given for each body segment, with the angle given by the VICON as the independent variable and the angle given by the DOTs as the dependent variable. For the Whole Body rolling and Leg Drop trials, the angle compared was rotation in the transverse plane around a vertical axis, as this is most relevant to sleep biomechanics (rolling over in bed to change position). For the Sitting Up trial, the angle compared was rotation in the sagittal plane around a frontal axis, as this angle indicates if a participant is sitting up or standing (and therefore awake and/or out of bed); only the sternum DOT data were used for the sitting up trial. Correlation coefficients, standard error, and confidence intervals were calculated for each DOT (sternum/pelvis/left thigh/right thigh), and static (at rest) and dynamic (in motion) phases were examined separately for all three movement trials. An error of <5° is accepted as being excellent for sensors of this kind [30], and so agreement of the DOTs orientation to within 5° of the corresponding Vicon orientation in both static and dynamic phases was considered feasible.

## 3. Results

All participants (n = 10) completed the overnight assessment. The distribution of gender was 70% male, with an overall age distribution of 20 to 59 years; baseline demographics are presented in Table 2. There were no adverse events for any participants.

### 3.1. Drift

In both trials, the degree of drift was minimal. In the moving test (using the hexagonal box test rig), there was negligible deviation from the expected angle (see Figure 3 for an example graph). In the stationary test, there was no deviation from the expected angle over 8 h of testing (see Figure 4 for an example graph). The maximum variation in outputted orientation was 0.15°; the difference between the first sample and last sample was 0.03°. These results demonstrate drift was <1° for >95% of the time, and therefore drift can be rated as Continue Without Modifications.

### 3.2. Battery Life

The average battery life overall was 9.06 h, with a range of 8.49 h to 9.84 h. This exceeded the predetermined threshold of 8 h, and also exceeded the average sleep duration of the participants (8.15 h). Battery life can therefore be rated as Continue Without Modifications.

### 3.3. Reliability of Recording

Recording of data proved to be highly reliable, with no drop out or missing data during any of the 10 trials. This is below the threshold of less than 5 s dropout, for <5% of the total recording time and so can be rated as Continue Without Modifications.

### 3.4. Comfort of Participants

Ratings of comfort were generally high (see Table 3), with an overall average satisfaction rating of 9.4 (8–10) and an overall average comfort rating of 9.5 (9–10). Common themes of qualitative feedback included the sensors not interfering with sleep, no issues with secure DOTs attachment, and occasional issues with sensors pairing with the app that were resolved with an additional attempt. There were some reports of discomfort removing the dressing in the morning, pulling on skin and hair, and causing mild erythema. Comfort of Participants was therefore rated as Continue But Monitor Closely.

### 3.5. Sensor Validity

The Xsens DOT platform and the VICON motion analysis lab demonstrated a high level of agreement. The greatest mean difference between systems was 0.025°. The standard error was <0.01° in all trials. For the movement evaluations, coefficients ranged from 0.9886 to 1.0081 for Whole Body rolling, 0.9741 to 1.0088 for Leg Drops, and was 1.0066 in the Sitting Up trial (only sternum DOT data used). For the stationary evaluations, coefficients ranged from 0.9929 to 1.0122 for supine and left and right side lying, 0.9886 to 1.0135 for Leg Drops, and was 0.9937 in the Sitting Up trial. Full results are presented in Table 4. There was an extremely small bias of the DOTs in underestimating the VICON reading (−0.0011°) within the recommended limit of <5° error [30].

## 4. Discussion

Overall, the results of this study suggest that the assessment of sleep biomechanics with Xsens DOTs can be considered feasible and valid in healthy adults. While considerations will have to be taken in future studies with different populations (children, those with physical and intellectual disabilities, etc.), it appears from the results of this study that the Xsens DOTs are fit for purpose in the recording of sleep biomechanics.

Drift was minimal in our testing of the DOTs, under both dynamic and static conditions—this concurs with findings of minimal drift in previous Xsens IMUs such as the Awinda [31] and MVN [32]. While this is likely to have been accounted for by internal quality assessment testing pre-launch, the absence of drift highlights one of the advantages of using IMUs over traditional accelerometry—the ability to integrate multiple data streams and maintain a consistent reading, as compared to the single data stream typical of accelerometers. By using a Kalman filter to combine the inputs of the accelerometer, magnetometer, and gyroscope components, any interference or deviation in one component can be compensated for by the other two—this is not possible in traditional accelerometry. Further, error from drift is cumulative and will aggregate over time; for example, a seemingly trivial drift error of 0.1°/min of recording will result in a 6° drift error after an hour of recording. Minimal drift over a 3.5 h trial is therefore very encouraging. Based on these results, it is therefore reasonable to proceed with a high level of confidence that drift will have minimal impact on future assessments of sleep biomechanics using the DOTs.

The battery life of the V2 Xsens DOTs proved to be sufficient for the recording of typical adult sleeping durations of 7–8 h [26]. The battery life of the current iteration of the DOTs can therefore be considered feasible in adults; however, future studies should determine feasibility in paediatric groups. Younger children (<10 years) are well acknowledged as having longer sleep durations (9–11 h) than adults [33], and so planning in future studies may have to account for longer recording times being required for feasibility. Extending battery life with an external power supply may be possible, but would likely negatively impact comfort and usability for participants—it may also pose a safety risk, for example, if a child participant removed an external battery. Other measures that may assist with data collection could include a delayed start (recording only the middle of a child’s sleep) or awaiting future hardware improvements from Xsens (the current V2 DOTS have a significantly improved battery capacity of 70 mAh, compared to the v1 45 mAh battery).

The DOTS proved to be very reliable when operating in recording mode. While logging data directly to the internal memory of the DOTs does bring about the limitation of the internal memory capacity of each DOT, a low enough data capture rate ensures that this capacity does not limit data collection length. A higher data capture rate is generally more desirable, as it gives a higher density and richness of data; however, for the recording of position as well as relatively slow and infrequent movements during sleep, a lower data capture rate (such as 4 Hz) is adequate. In addition, the use of recording mode avoids issues with remaining in the Bluetooth transmission range of the receiving smartphone, which may cause dropout and data loss if participants move away from it (e.g., for a bathroom break during the night). This is important as future studies are likely to be in groups whose adherence to instructions is likely to be low (children and those with intellectual disabilities), so removing the need for a smartphone to be carried around ensures the assessment is far easier to implement.

The comfort of participants was demonstrated as being feasible for future studies, with participants indicating that they were generally comfortable wearing, sleeping with, and operating the DOTs. There were occasional reports of discomfort and some erythema when removing the attachment dressings. While this did not result in any significant problems for any participants, it must be monitored for and taken into account in future studies as it may result in issues with compliance. The assessment of participant comfort highlights a number of limitations with this study. Participants were recruited via convenience sampling and so may have been predisposed to give positive responses when questioned about comfort during sleeping while wearing the DOTs. Further, all participants in this study were healthy adults; future studies with the DOTs could include both typically developing children and children with disabilities. Lower compliance common in children (especially considering pain or discomfort removing dressings) or those with intellectual disabilities (or a combination of both) are likely to raise issues with comfort/tolerance of wearing the DOTs during sleep in these groups, and further problem solving may be needed.

The VICON motion capture system and the Xsens DOTs platform demonstrated a very high level of agreement in the measurement of orientation in all trials, both in motion (transverse rotation/rolling) and at rest. With correlation coefficients of >0.9 and error of less than <5° considered as excellent [30], the DOTs can be considered as having performed extremely accurately. While there was a bias present, it is extremely small (<0.001°) and so will not functionally affect the recording of sleep biomechanics. The DOTs can therefore be considered valid and be used with confidence in future applications of this kind.

We acknowledge a number of limitations of this study. With respect to the Battery, Drift, and Reliability metrics, these were examined in a benchtop setting—there may be variation in situ when used to assess real sleeping individuals in their native sleep environment, and so future studies should monitor for this possibility. The VICON sub-study had a small sample size (n = 5); however, due to the repeated movements during trials, the low amplitude of the movements, and the strength of the correlations found in the current data set, we are confident that the chance of type 2 error is low. Another limitation was the inability to examine the level of agreement between the Xsens DOT and VICON systems in prone, as adopting this posture obscures the retroreflective markers and renders the VICON system unable to record this position. However, based on the high performance of the Xsens DOTs in all other positions, it is highly unlikely that findings would be different when participants would be in the prone position. Relatedly, this experiment was performed using simulated sleep movements by awake participants rather than actual sleeping participants. While capturing movement during sleep would be preferable, it was determined to be impractical to have participants sleep inside the Motion Capture Laboratory, and simulated movements were considered an acceptable substitute as they are comparable in rate and range of motion. Finally, VICON testing took place in a controlled, magnetically stable environment—future use in the home environment may have differing magnetic fields and ferrous objects (such as computers, iron bedframes, etc.) that can impact the accuracy of the magnetometer component of the IMUs. However, this risk can be managed with the magnetic field mapping function of the Xsens DOT app to calibrate data collection for each new magnetic environment.

## Figures and Tables

**Figure 1 biomimetics-08-00002-f001:**
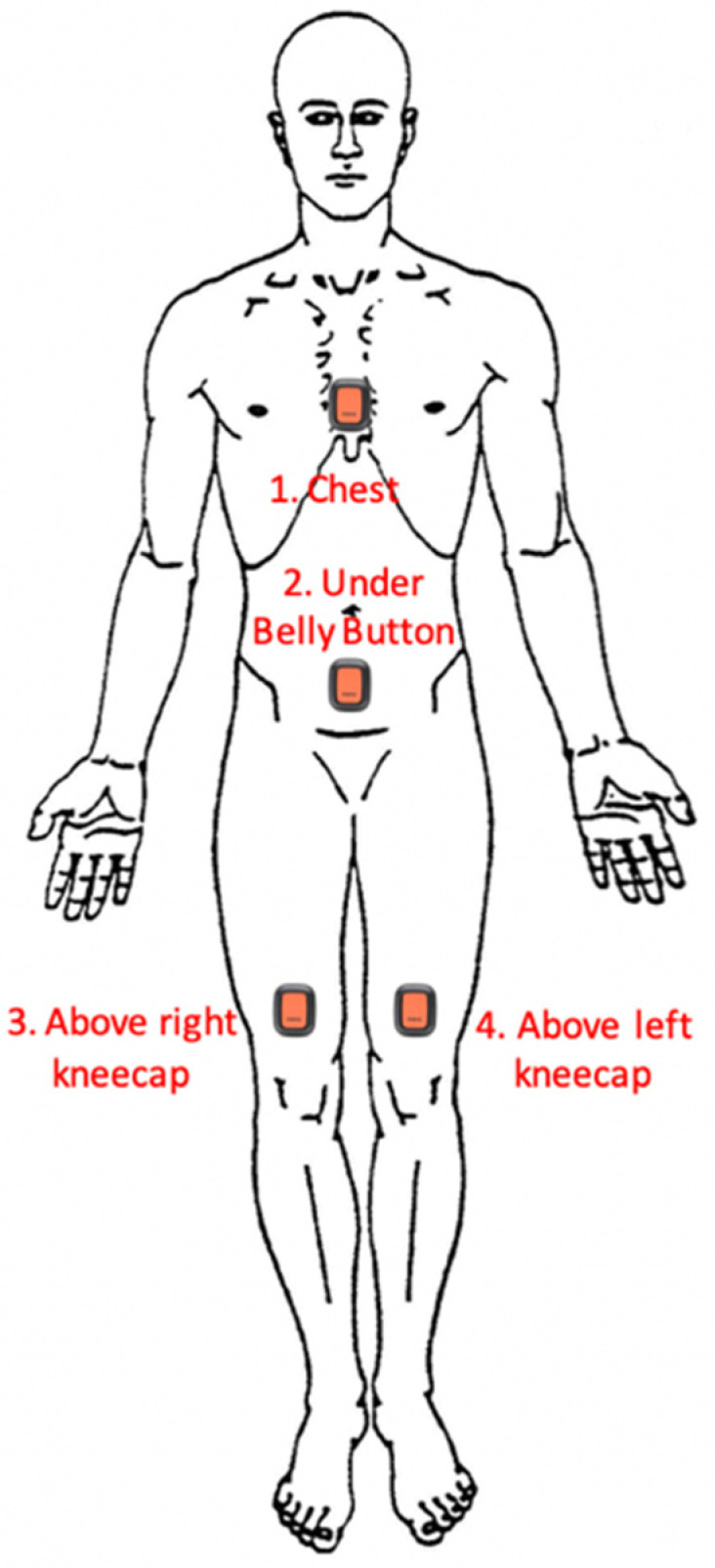
Locations of Xsens DOT Placement on participants.

**Figure 2 biomimetics-08-00002-f002:**
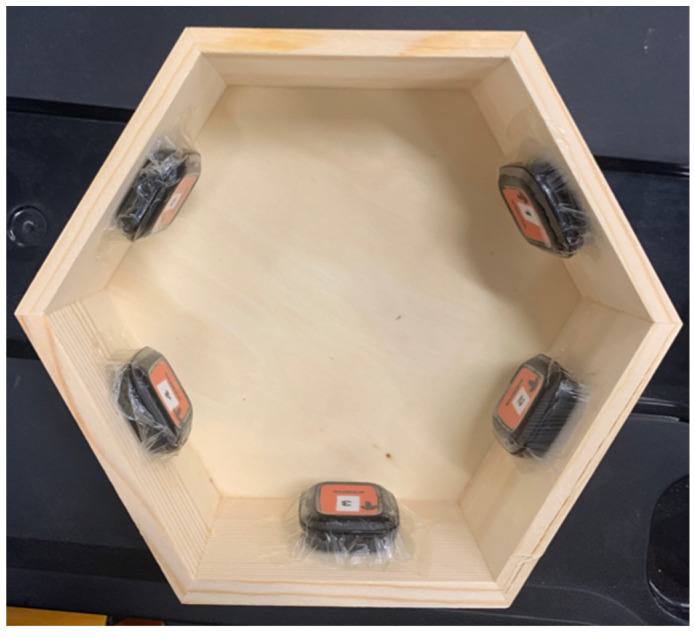
Apparatus for testing of drift under dynamic conditions.

**Figure 3 biomimetics-08-00002-f003:**
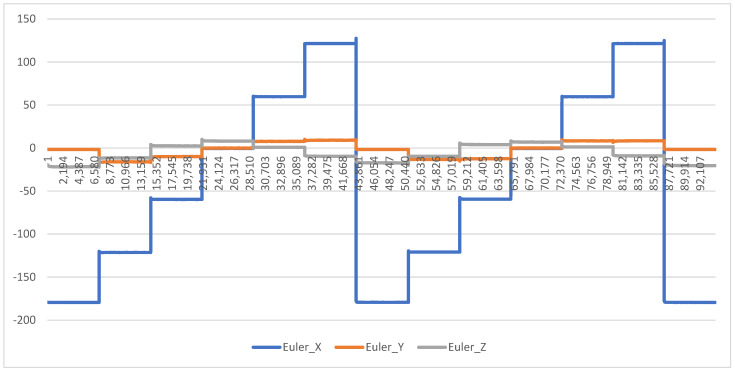
Orientation (in degrees) during Dynamic Drift Testing, demonstrating the expected 60-degree changes in output orientation at 30 min intervals.

**Figure 4 biomimetics-08-00002-f004:**
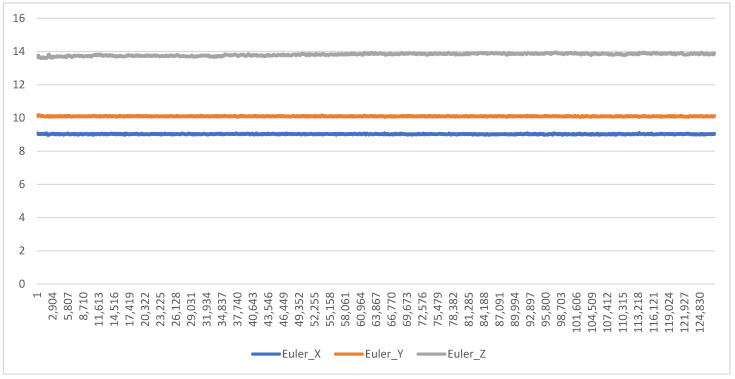
Orientation (in degrees) during Static Drift Testing, demonstrating no change in orientation during an 8 h recording.

**Table 1 biomimetics-08-00002-t001:** Feasibility criteria according to Thabane et al [25].

	Accepted Feasibility Outcomes	Possible Feasibility Outcomes
Drift	The Xsens DOTs remain within 1 degree of expected reading >95% of the time and do not drift	Continue without modifications (*feasible as is)*
Battery Life	The Xsens DOTs battery life can allow for 8 h of continuous recording	Continue without modifications, but monitor closely (*feasible with close monitoring)*
Reliability	The Xsens DOTs can reliably stream data >95% of the time, with maximum dropout periods of <5 s	Continue, but modify protocol (*feasible with modifications)*
Wearer Comfort	>90% of participants give a rating of >7/10 for comfort while sleeping wearing the Xsens DOT	Stop (*main study not feasible)*

**Table 2 biomimetics-08-00002-t002:** Baseline demographic data of study participants.

Gender	7 M/3F
Age	20 to 59 years, mean 29.4 (11.1 SD)
Weight	54 to 91 kg, mean 69.8 (11.1 SD)
Height	166 to 195 cm, mean 175.9 (10.5 SD)

**Table 3 biomimetics-08-00002-t003:** Results of Participant Comfort Survey.

Question	Average Score
Were the instructions clear?	9.8
Was the app easy to navigate?	9.4
Was the equipment easy to use?	9.5
How much did the sensors interfere with your sleep?	9.5
How well did the sensors stay attached?	10.0
Overall, how satisfied were you with using the sensors?	9.4
Overall, how comfortable were the sensors while you were sleeping?	9.5

**Table 4 biomimetics-08-00002-t004:** Results of VICON Motion Analysis Lab Testing (Dynamic and Static).

Dynamic Phase						
**Whole Body Rolling (Mean Rotation)**						
	**Coef.**	**Std. Err.**	**z**	**P > z**	**[95% Conf. Interval]**	
Sternum	0.9922	0.0037	268.12	<0.001	0.9849	0.9995
Pelvis	0.9886	0.0033	301.02	<0.001	0.9822	0.9950
Left Thigh	1.0081	0.0018	555.41	<0.001	1.0046	1.0117
Right Thigh	0.9973	0.0031	323.9	<0.001	0.9913	1.0033
**Leg Drops (Mean Rotation)**						
	**Coef.**	**Std. Err.**	**z**	**P > z**	**[95% Conf. Interval]**	
Sternum	1.0039	0.0063	158.26	<0.001	0.9915	1.0163
Pelvis	0.9741	0.0036	268.5	<0.001	0.9669	0.9812
Left Thigh	1.0088	0.0021	489.14	<0.001	1.0048	1.0129
Right Thigh	1.0017	0.0030	331.49	<0.001	0.9958	1.0076
**Sitting Up**						
	**Coef.**	**Std. Err.**	**z**	**P > z**	**[95% Conf. Interval]**	
Sternum	1.0066	0.0042	240.72	<0.001	0.9984	1.0148
**Static Phase**						
**Whole Body Rolling (Mean Rotation)**						
	**Coef.**	**Std. Err.**	**z**	**P > z**	**[95% Conf. Interval]**	
Sternum	0.9929	0.0030	333.92	0	0.9871	0.9987
Pelvis	0.9917	0.0017	593.43	0	0.9884	0.9950
Left Thigh	1.0122	0.0034	295.61	0	1.0055	1.0189
Right Thigh	0.9983	0.0030	329.76	0	0.9923	1.0042
**Leg Drops (Mean Rotation)**						
	**Coef.**	**Std. Err.**	**z**	**P > z**	**[95% Conf. Interval]**	
Sternum	1.0135	0.0066	153	0	1.0005	1.0265
Pelvis	0.9886	0.0023	427.36	0	0.9841	0.9931
Left Thigh	1.0094	0.0025	409.85	0	1.0045	1.0142
Right Thigh	0.9992	0.0040	246.89	0	0.9913	1.0072
**Sitting Up**						
	**Coef.**	**Std. Err.**	**z**	**P > z**	**[95% Conf. Interval]**	
Sternum	0.9937	0.0014	708.3	0	0.9910	0.9965

## Data Availability

All authors had access to the study data.

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
