# Peer review of "Can Wearable Inertial Measurement Units Be Used to Measure Sleep Biomechanics? Establishing Initial Feasibility and Validity"

_biomimetics, 2022, doi:10.3390/biomimetics8010002_

Round 1

Reviewer 1 Report

Authors presented a well designed research on measurement of sleeping biomechanics with IMU wearable sensors. This study was compared against the 'gold standard' motion capture system. However, one key concern for me is that how the sensors were fixed, and if the placement would affect the perception of comfort during sleeping? 

Reviewer 2 Report

In the manuscript “Can wearable Inertial Measurement Units be used to measure sleep biomechanics? Establishing initial feasibility and validity” the authors described a study in which they assess the feasibility of Xsens DOT platform for recording sleep biomechanics, through four metrics: Drift, Battery Life, Reliability of Recording, and Participant Comfort. Then, they evaluated the validity of this system by comparing the output with a VICON motion capture system. The results revealed that Xsens DOTs are able to record sleep biomechanics. Furthermore VICON motion capture system and the Xsens DOTs platform showed a high level of agreement in the measurement of orientation in all trials, both in motion (transverse rotation/rolling) and at rest. This paper could represent a technological progress toward a more ecologic assessment of sleep biomechanics.

The study is well structured, interesting and of value to a range of fields. Nonetheless, I have some concerns that should be addressed.
The metrics used to analyze the feasibility of the method were not studied in a real sleeping subject (except for Participant Comfort), and this did not allow for a study in an ecological system. Metrics may reflect different measurements in real sleeping subjects. On the same line, the comparison between VICON and DOTs was carried out on simulated sleeping movements performed by awake subjects.
Also, how did you compare the DOTs system, that samples at 4Hz, with VICON system, that samples at 250Hz?
In the introduction, in the part where you present the second aim, there are no insights about motion analysis systems used to study sleep biomechanics and some citation are needed too.
As regards Table 1, I suggest inserting the possible outcomes of each evaluation metric, with their respective descriptions.
In Results, drift was <1° for 95% of the time, but, being a cumulative measures, how much was the final drift error?
In Discussion, regarding the Participant Comfort, it is not mention the discomfort in removing the dressing (causing mild erythema).
Line 157: As its proven that Bluetooth is unreliable, I expected some reference to support this declaration in this line.
Figure 3.:The graph is unclear. I also suggest to add some description.
